# A 1083 nm Narrow-Linewidth DFB Semiconductor Laser for Quantum Magnetometry

Mengying Wu [1,2], Haiyang Yu [1,2], Wenyu Wang [1,2], Shaojie Li [1,2], Yulian Cao [1,2,*] and Jianguo Liu [1,2,*]

1 State Key Laboratory of Integrated Optoelectronics, Institute of Semiconductors, Chinese Academy of Sciences, Beijing 100083, China; wumengying123@semi.ac.cn (M.W.); hyyu@semi.ac.cn (H.Y.); wangwenyu@semi.ac.cn (W.W.); lishaojie@semi.ac.cn (S.L.)
2 College of Electronic, Electrical and Communication Engineering, University of Chinese Academy of Sciences, Beijing 100083, China
* Correspondence: caoyl@semi.ac.cn (Y.C.); jgliu@semi.ac.cn (J.L.)

**Abstract:** A 1083 nm laser, corresponding to a characteristic spectral line of $^3$He $2^3S_1$-$2^3P$, is the core light source for spin-exchange optical pumping-free technology, and thus has important developmental significance. In this paper, precise wavelength 1083.34 nm semiconductor lasers with 285 mW output power, $-144.73$ dBc/Hz RIN noise and 30.9952 kHz linewidth have been successfully achieved via reasonable chips design, high-quality epitaxial growth process and ultra-low reflectivity coating fabrication. All the results show the highest output power and ultra-narrow linewidth of the single-frequency 1083 nm DFB semiconductor laser achieved in this paper, which can fully satisfy the requirement of quantum magnetometers.

**Keywords:** semiconductor lasers; magnetometers; 1083 nm



## 1. Introduction

Brain science is recognized as the most valuable scientific research in the 21st century [1]. There are three main directions of brain science: the construction of basic neuron structures [2], the understanding of the coordination approach to cognitive behavior in individual brain areas [3], and the development of artificial intelligence and brain–computer interfaces via imitating human brain activity patterns [4–6]. All such studies are based on the clarification of the brain structure [7]. Compared to the traditional electroencephalogram method, the magnetoencephalogram (MEG) has become the superior choice to clarifying the brain structure by eliminating the usage of liquid conductive silicone; it also has advantages such as a wide test range, non-radiation, non-trauma and non-invasion of the human body [8–12]. The brain magnetic field intensity is only about 100 fT, which requires a highly sensitive detector [13]. Meanwhile, position accuracy in MEG also limits the size of the detector [14]. As a result, it is important to develop an ultra-high-sensitivity detector in a compact size for MEG. Spin-exchange relaxation-free (SERF) technology helps to reduce the volume and significantly promotes the sensitivity and spatial resolution of detectors, and it has achieved the highest sensitivity, 0.16 fT/$\sqrt{\text{Hz}}$, in weak magnetic detection [15]. The $^3$He atom is an important detection-sensing medium in SERF because of its metastable structure without a nuclear spin [16].

A 1083 nm laser, corresponding to a characteristic spectral line of $^3$He $2^3S_1$-$2^3P$, is a core light source for SERF [17]. Solid-state lasers and fiber lasers are two typical types of 1083 nm lasers. Traditional lasers need to dope rare earth elements such as neodymium (Nd) or yttrium (Yb) in order to achieve precise emissions at a 1083 nm wavelength, which is not only expensive but also requires additional laser pumping [18,19]. Until now, all available lasers have been unable to meet the high-position-accuracy requirement of a MEG detector due to bulky volume [20]. A 1083 nm semiconductor laser has the advantages of small size, low cost, simple structure and high stability; therefore, it is the most suitable

laser for SERF of MEG detector [21,22]. However, there are few reports on semiconductor lasers with a linewidth of less than 1 MHz and a lasing of around 1 μm. A narrow linewidth stands for low noise, which is important for a MEG to detect a weak single. In this paper, a 1083 nm InGaAs/GaAsP quantum well semiconductor laser with excellent performance is prepared. The lasing wavelength is precisely controlled at 1083.34 nm, the output power is 285 mW, the side-mode suppression ratio (SMSR) is greater than 45 dB, the linewidth is 30.9952 kHz and the RIN noise reaches $-144.73$ dBc/Hz; these examples of excellent performances drive the further development of a magnetoencephalogram.

## 2. Laser Design Simulation and Fabrication

InGaAs is selected as the quantum well material to achieve 1083 nm. Indium with a larger size can strongly hinder the propagation of epitaxy defects and effectively suppress the growth rate of dark line defects [23]. InGaAs and GaAs may have a lattice distortion greater than 2% to achieve a 1 μm light emission, and dislocation defects are prone to occur in this case [24]. In order to prevent strain accumulation, GaAsP with tensile strain characteristics is selected as the barrier material in this paper [25]. The indium composition of the quantum well is 35%, and the thickness of the quantum well is 6 nm. The phosphorus compositions of the barrier are 10%, and the thickness of the barrier is 10 nm. A triple quantum well is chosen in order to obtain a higher gain [26].

Reducing the loss to improve the Q factor is an effective method for narrowing the linewidth of DFB lasers. Free-carrier losses and leakage losses are the two main loss resources for 1 μm emission wavelength lasers. Almost all free carriers come from the cladding layer. In order to reduce the free-carrier loss [27], the waveguide layer thickness could be increased to minimize the overlap integral between the optical field and the cladding layer. However, the leakage loss is directly proportional to the waveguide thickness [28,29]. Therefore, it is a trade-off between leakage loss and free-carrier loss. Commercially available laser-diode simulators, PIC3D and Lumerical, are applied to simulate the behavior of a designed laser diode. As shown in Figure 1a, with the thickness of the waveguide increasing, the output power first increases and then decreases. Combining the relationship between the fundamental-mode confinement factor and the waveguide thickness in Figure 1b, it can be seen that the output power is mainly affected by the free-carrier loss when the waveguide layer is thin. With the waveguide thickness increasing, free-carrier loss decreases and output power increases. When the waveguide thickness reaches 400 nm, the output power is mainly affected by the confinement factor, which reflects a leakage loss. The confinement factor decreases as the waveguide layer becomes thicker, proving that the leakage loss increases with the waveguide thickness increase; as a result, the output power decreases at the same time [30]. In this paper, InGaAs/GaAsP QWs were sandwiched between two 400 nm $Al_{0.08}GaAs$ waveguide layers. The sandwich structure is designed to obtain the maximum output power and the minimum total loss. A thick waveguide layer also can decrease the far-field divergence angle, which is beneficial to releasing the impact of spatial hole burning effects and improving operation stability [31]. The 1250 nm-thick cladding layer p-$Al_{0.3}GaAs$ gradually changes doping, and the doping concentration increases from $1 \times 10^{18}$ cm$^{-3}$ to $2 \times 10^{18}$ cm$^{-3}$ with the distance from the MQWs increasing. The gradually changing doping not only ensures a small series resistance but also reduces the free-carrier loss [32]. A 200 nm highly p-doped GaAs contact layer is grown on top of the structure to improve the metal–semiconductor contact.

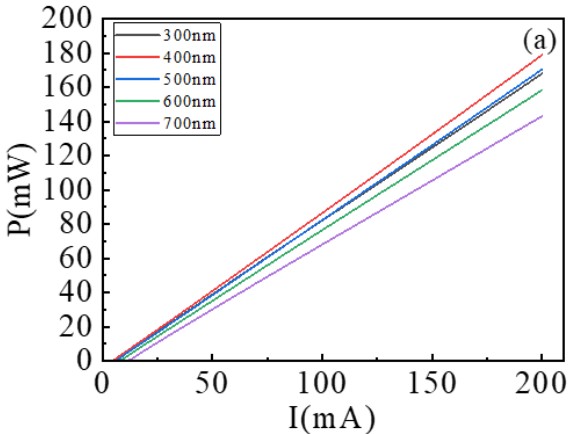
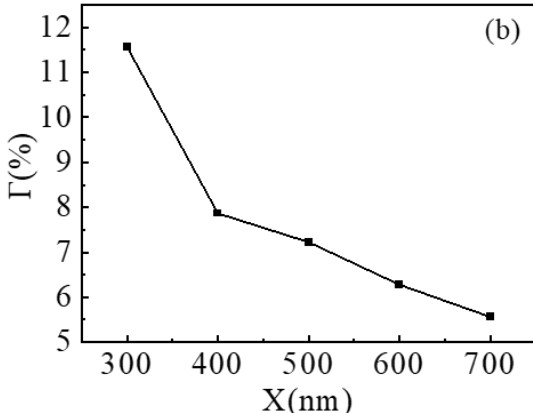

**Figure 1.** (**a**) The relationship between output power and current at different waveguide thickness (simulated by PIC3D). (**b**) Variation of fundamental mode confinement factor with different waveguide thickness (simulated by Lumerical).

The premise of the narrow linewidth is a single-mode emission [33]. The coupling coefficient can be modified by altering the GaAs/InGaP/GaAs grating structure to ensure a single-longitudinal mode emission and a low loss [34]. Considering the requirements of manufacturing precision, second-order grating is selected in this paper. The period of grating is 316 nm. The thickness of the grating etching layer is 200 nm, the thickness of GaAs layer under the grating etching layer is 50 nm, and the thickness of the GaAs grating cover layer is 50 nm. Figure 2 exhibits the coupling coefficient decreasing with the increase in the distance between the grating layer and the MQWs at different etching depths. The point in the black shadow represents the structures with a coupling coefficient of no more than 10 cm$^{-1}$. This range can guarantee single-mode stability without introducing excessive loss, when the cavity length is 1000–2000 μm [35]. It can be seen from Figure 3 that the output power and the SMSR gradually increase with the increasing etching depth as the distance between the grating layer and the MQWs is 650 nm. When the etching depth reaches 40 nm, that is, the coupling coefficient is greater than 10 cm$^{-1}$, the output power decreases and the SMSR hardly increases. In order to ensure a low loss, which is beneficial to improve the Q factor, a single-longitudinal mode, a 30 nm etching depth and a 650 nm distance are selected in this paper. The coupling coefficient is 8 cm$^{-1}$. A high Q factor and a single mode can effectively ensure narrow linewidth. In order to avoid the spatial hole-burning effect, the grating with four λ/16 phase-shifted regions is adopted to ensure a high-fundamental-mode output power. The spatial hole-burning effect causing mode competition can lead to a gain in saturation at the lasing wavelength. The carriers used for side mode gain increase, and those used for lasing wavelength gain decrease, which is equivalent to increasing the loss and reducing the Q factor. This will broaden the linewidth [36,37]. In order to ensure the stability of the single-transverse mode, the shape of the double-trench-ridge waveguide is optimized to increase the leakage loss of the higher-order lateral modes [36]. The structure is set with a ridge width of 3 μm and a groove width of 9 μm. In Figure 4, the simulated curves, respectively, represent the change trend of confinement factors of the fundamental mode and the first-order lateral mode with the distance between the stop position of the ridge waveguide and the grating layer. The confinement factor of the first-order lateral mode decreases with increase in the distance. The confinement of the fundamental mode is stable. The maximum confinement-factor difference of two modes is achieved at a 75 nm distance, which is most conducive to single-transverse mode emission. At the same time, the fundamental mode confinement factor can reach a high value, 7.13%, at a 75 nm distance. In order to guarantee a single-transverse mode emission and a high-fundamental-mode gain, 75 nm is the optimal spacing between the ridge waveguide stop position and the grating layer. Therefore, 1125 nm is the optimal

etching depth for the ridge waveguide. The final chip structure diagram is shown in Figure 5.

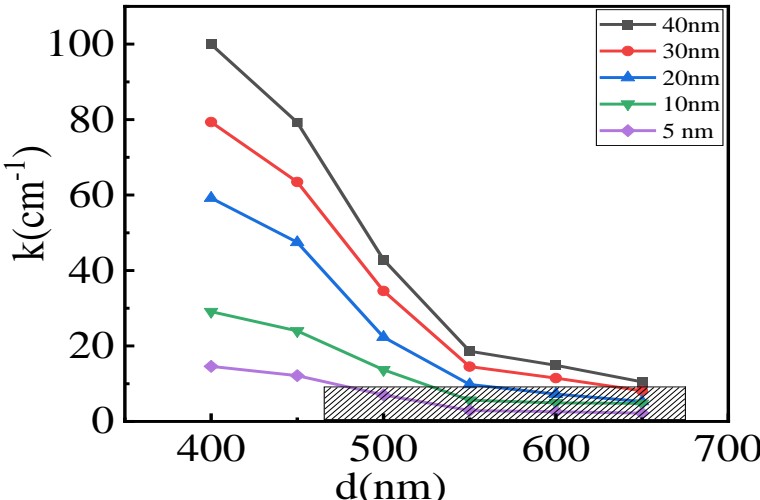

**Figure 2.** The variation trend of the coupling coefficient with the distance between grating layer and MQWs at different grating etching depths (simulated by PIC3D).

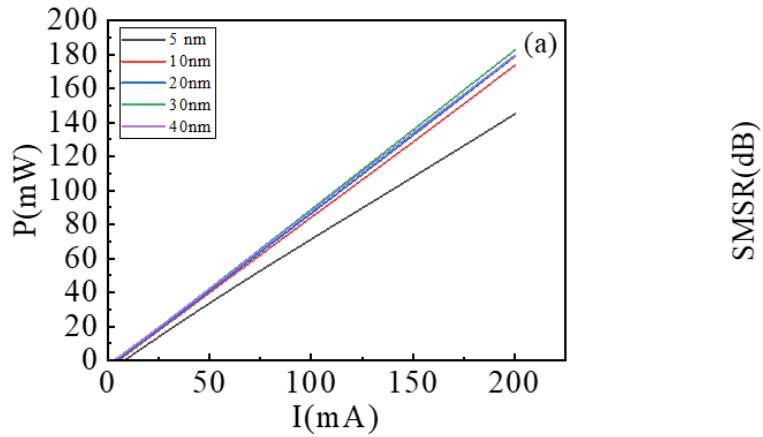

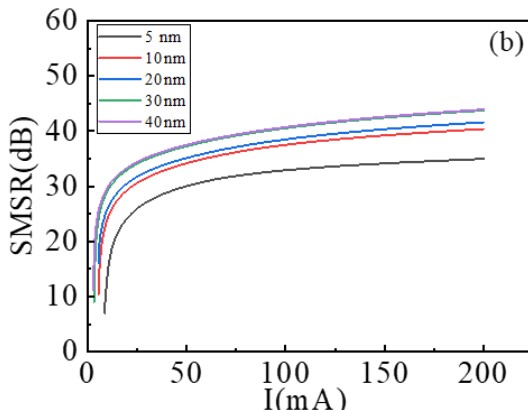

**Figure 3.** (**a**) The relationship between output power and current at different grating etching depths (simulated by PIC3D). (**b**) The relationship between SMSR and current at different grating etching depths (simulated by PIC3D).

The structure was grown via metal–organic chemical-vapor deposition (MOCVD). The growth temperature was increased to obtain a lower background doping concentration when growing the undoped waveguide layers, which can reduce the free-carrier loss and improve the Q factor. The interface quality is the most important index of the quantum well layers. According to experimental optimization, the growth temperature of 565 °C and a high V/III group proportion ratio were employed for the quantum well growth. The routine phosphorus source and arsenic source used in the GaAsP epitaxial growth process are their full hydrogen compounds, that is, phosphine ($PH_3$) and arsine ($AsH_3$). The decomposition temperature of $PH_3$ and $AsH_3$ is relatively high, which is not conducive to the low-temperature growth of InGaAs. At the same time, because the thermal stability of $AsH_3$ is significantly worse than that of $PH_3$, the control of the element content of the epitaxial material is more difficult. In this work, tertiarybulphosphine (TBP) and tertiarybularsine (TBAs) had a low decomposed temperature and were replaced by $PH_3$ and $AsH_3$, which was beneficial to InGaAs epitaxial growth. In addition, the decomposition temperature of TBP and TBAs were similar, so the element content in the epitaxial layer could be more precisely controlled. It was conducive to improving the uniformity and

the repeatability of epitaxial materials, reducing the loss introduced during growth, and improving the Q factor. During the etching grating process, the dry-etching and the wet-etching techniques were combined to obtain a smooth and high-consistency surface; such an etching method can also improve the epilayer quality and the Q factor. A scanning electron microscope (SEM) image of grating is presented in Figure 6a. The ridge stripe with a width of about 3 μm and the trench with a width of about 9 μm were processed with dry-etching techniques using the SiO2 as a mask. An SEM image of the optimized ridge waveguide is presented in Figure 6b. The cavity length is 1000 μm. An antireflective film with 0.01% reflectivity at 1000–1200 nm coated on the front facet is used to eliminate the mode selection effect of the cavity. The relationship between wavelength and reflectivity is shown in Figure 7. The detailed parameters of the antireflective film structure are listed in Table 1. The lasing mode is filtered by the Bragg mode selection conditions, and carriers are avoided from exhausting at the side mode gain, which reduces the loss and improves the Q factor [36,37]. A highly reflective film with 99% reflectivity was coated onto the rear facet.

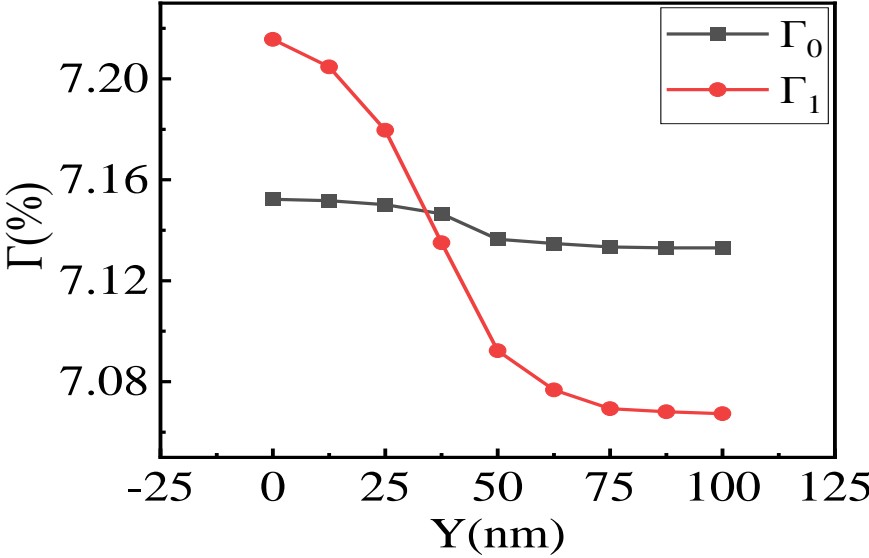

**Figure 4.** The variation trend of confinement factors of fundamental mode and first-order lateral mode with the distance between stop position of ridge waveguide and grating layer (simulated by Lumerical).

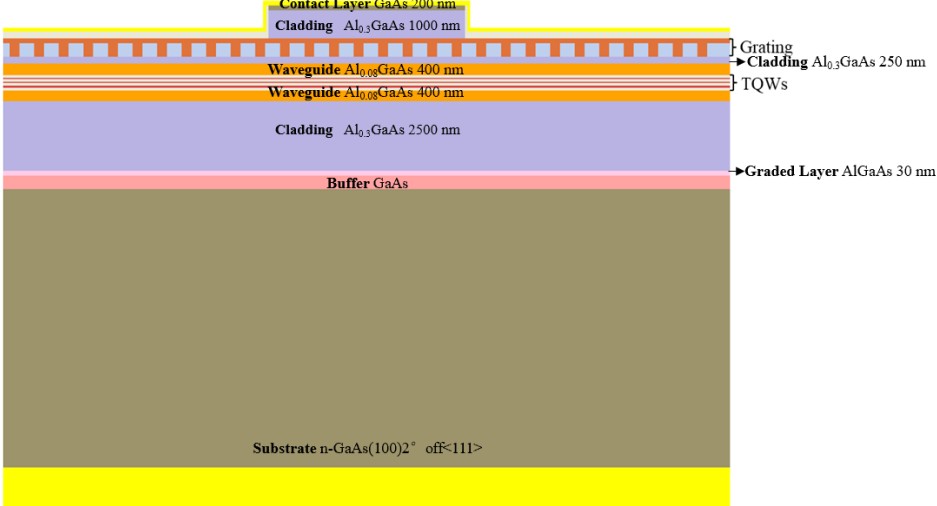

**Figure 5.** The chip structure diagram.

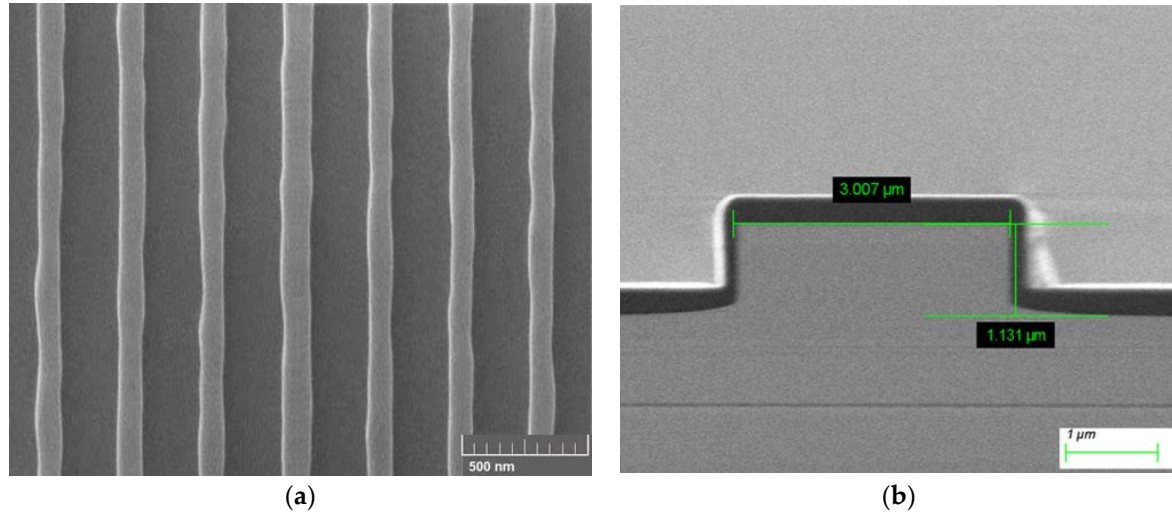

**Figure 6.** SEM image (**a**) grating (**b**) ridge waveguide.

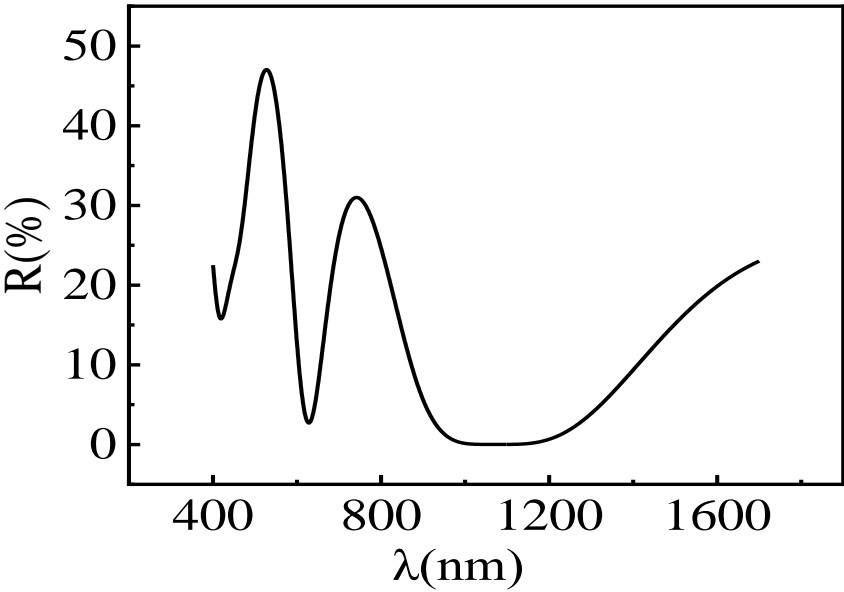

**Figure 7.** The relationship between wavelength and reflectivity.

**Table 1.** The antireflective film structure.

| Layer | Thickness (nm) |
|:---:|:---:|
| $SiO_2$ | 235.87 |
| $Nb_2O_5$ | 42.98 |
| $SiO_2$ | 246.43 |
| $Nb_2O_5$ | 117.57 |

The encapsulation of the laser uses butterfly packaging that integrates the thermistor and the thermal electronic cooler (TEC). The material of the laser butterfly package is kovar alloy, and the dimensions of the butterfly package are 30 mm × 12.8 mm × 11 mm. The tungsten copper W60 with low tungsten content is selected as the heat sink material in order to increase the infiltration of tin during the TEC welding process. The packaging adopts the two-stage lens packaging method. In order to improve the isolation and reduce the insertion loss, an online isolator with 30 dB isolation and 1.1 dB insertion loss is used in

the paper. The pigtail output is adopted in the package, and the end surface of the fiber has an oblique 8° angle to prevent reflection. The pigtail is soldered and fixed at the position where both the coupling efficiency and the output power reach the maximum value. The packaging schematic is shown in Figure 8.

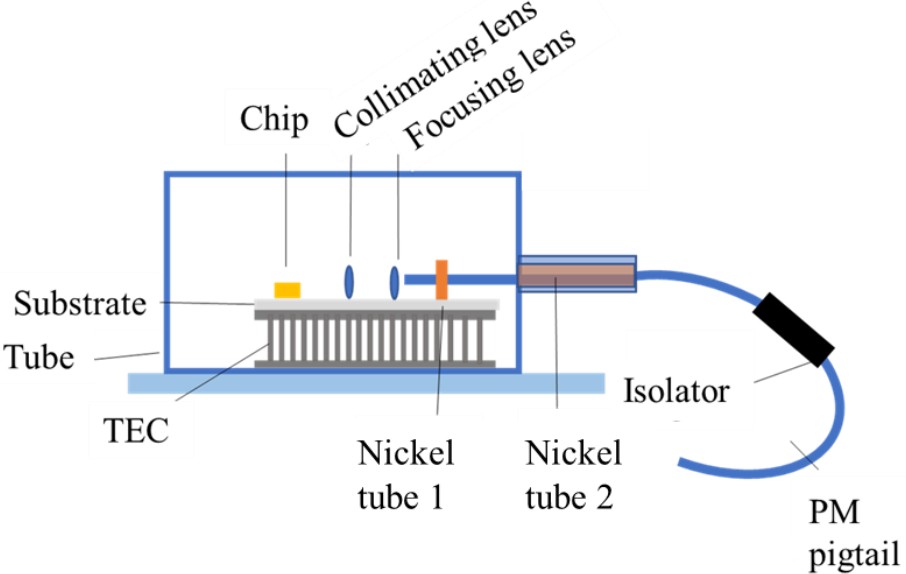

**Figure 8.** The packaging schematic.

## 3. Results

The chip preparation and the module packaging are completed according to the above design scheme. The light-current-voltage (LIV) measurements shown in Figure 9a reveal the continuous wave (CW) output power 285 mW at 500 mA in the single-mode operation, showing the highest output power among the reports that achieve a single-frequency 1083 nm DFB semiconductor laser [38,39]. This is attributed to the high gain achieved by the triple quantum well and the optimized ridge waveguide and the low loss obtained via the specified waveguide thickness, the adequate grating fabrication and the epitaxial growth conditions. The threshold current is 25 mA, and the slope efficiency ∼0.676 W/A. [40]. The light-current (LI) curve has no kinks or mode hop jumps from the threshold up to the saturation. The mode stability is accomplished via the employment of multiphase-shift grating which significantly suppresses mode competition, and the antireflection coating with 0.01% reflectivity which reduces the probability of side modes lasing introduced by cavity. The output power of the fabricated device is up to 30 mW at 100 mA and the coupling efficiency of packaging can reach 61%. Driving a conventional semiconductor laser with a current pulse can accomplish the gain switching [41]. The short cavity length, 1 mm, can ensure a short photon lifetime and achieve a short optical pulse width [42]. The high-fundamental-mode confinement factor, 7.13%, ensures a high peak photon density and a low gain suppression, which can also promise a short optical pulse width. At the same time, the high-fundamental-mode confinement factor ensures a high peak inversion level of the optical pulse [42]. Figure 9b shows the measured spectrum of the chip under a 200 mA injection current that sweeps the range from 1081 nm to 1085 nm. The lasing wavelengths are 1083.34 nm, which is almost coincident with the Bragg wavelength of the grating. It indicates that the SMSR is 48 dB, which is roughly coincident with the simulation result in Figure 3b. In order to protect the spectrum analyzer, the spectrum was measured after a laser attenuation, which reduces the contrast between the laser signal and the noise, showing the noise background.

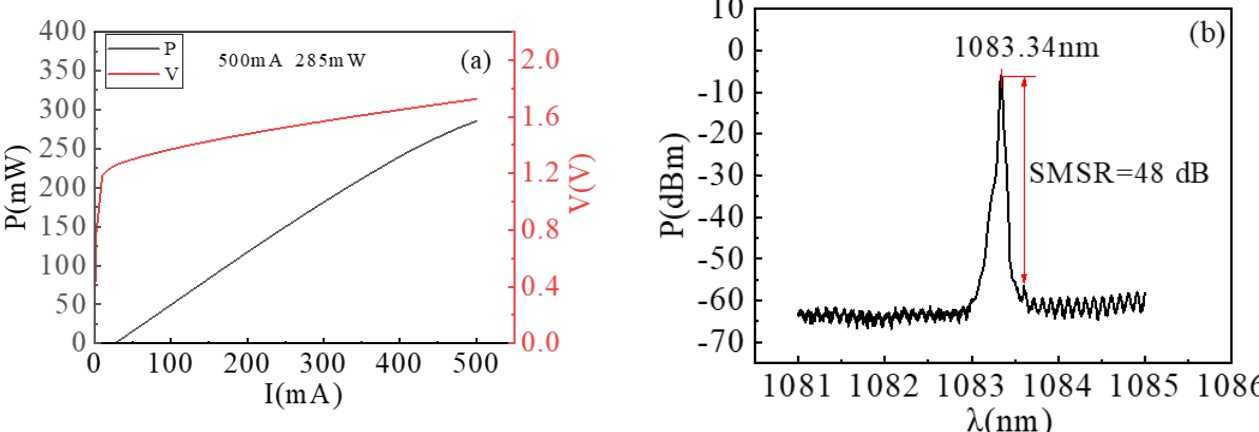

**Figure 9.** (**a**) Light-current (L-I) and voltage-current (I-V) characteristics of chip (**b**) spectrum.

To research the tuning accuracy and the tuning range of wavelength, we measured the spectra at different injection currents and different temperatures. As the injection current increases, the band gap of the semiconductor material decreases, the junction temperature of the heterojunction increases, and the gain spectrum of the quantum well is a red shift, these effects will lead to a red shift in the peak wavelength. It can be seen from Figure 10a that the peak wavelength is smoothly red shift with the injection current increasing and no obvious mode hopping is observed at 25 °C. We started to test the spectrum after the temperature stabilized at 25 °C and the current stabilized at the set value. The injection current increases by 100 mA, the peak wavelength increases by 0.3 nm, and the tuning accuracy can reach 0.003 nm/mA realizing precise wavelength control. This is very important for the MEG requiring an extremely high wavelength accuracy. By directly changing the junction temperature of the heterojunction, the red shift of the gain spectrum is faster. Figure 10b shows the spectrums of the chip at an injected current of 200 mA at different temperatures. We started to test the spectrums after the temperature stabilized at the set value and the current stabilized at 200 mA. The result indicates that the tunable range can achieve up to 2.4 nm with 0.08 nm/°C average change rate during 25–55 °C without mode hopping. A wider tuning range can be achieved via further modifying temperature and current.

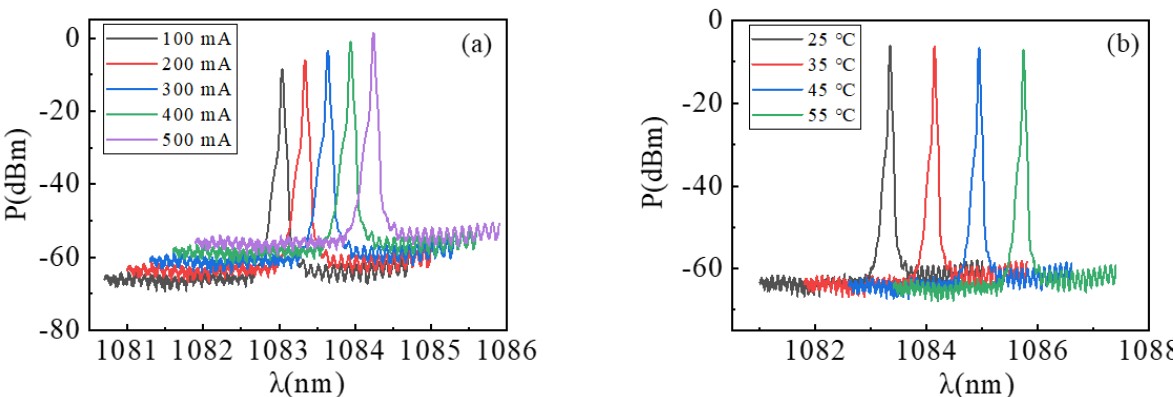

**Figure 10.** (**a**) Spectra at different injection currents at 25 °C. (**b**) Spectra at different temperature at 200 mA.

To obtain the accurate linewidth, we used two methods which were the self-delay heterodyne (SDH) method [43] and the power spectral density (PSD) of frequency noise method to measure [44,45]. In the SDH method, one incident light was delayed by a 10 km fiber line and the other beam as the signal beam shifted frequency to 200 MHz by an acoustic optic modulator. Then, the two lights were interfered with using Mach–Zehnder interferometer in a coupler and detected by a detector. As shown in Figure 11a, it is the signal of linewidth by the SDH method as the black line under 200 mA pump current. The red line shows the Lorentz fit of lasing signal with a 20 dB linewidth is 1.919 MHz, which indicates the measured laser FWHM linewidth is approximately 96 KHz. Using the PSD of frequency noise to calculate linewidth, we inputted the laser into the OEWAVE equipment OE4000, which measured the PSD of frequency noise, and obtained the linewidth through the β estimate method [46]. As shown in Figure 11b, the PSD of frequency noise is flat on the high-frequency side, whereas it increases the lower frequency. The calculated linewidth from the PSD of frequency noise by OE4000 is about 30.9952 KHz. The Lorentz-fitted linewidth measured by the delayed SDH method is limited by the noise model and only includes the contribution of the white noise components [47]. But the linewidth measured by the SDH method is larger than the value calculated by the PSD of frequency noise. This is because the length or the index of the very long fiber configuring the interferometric arm in the SDH method changes randomly because of temperature fluctuations, vibrations, and other types of environmental disturbances, thus it induces low-frequency random phase drifts in the interferometric signal [48]. At the same time, the tail of the spectrum measured by the SDH method is not taken into consideration in the Lorentz fitting process, resulting in the fitted value of white noise larger than the real value [44]. Therefore, the PSD of the frequency noise method is recommended to completely describe the frequency noise behavior and the linewidth.

We obtain the dependence of the FWHM linewidth calculated by the PSD of frequency noise on the output power depicted in Figure 11c. In general, the theoretical expected spectral linewidth of the single-mode semiconductor lasers caused by the spontaneous emission and the carrier concentration noise can be calculated by [48–51]:

$$\Delta v = \left(1 + \alpha^2\right)\frac{R_{sp}}{4\pi I} \tag{1}$$

where $R_{sp}$ is the rate of spontaneous emission into the lasing mode, $\alpha$ is the linewidth-broadening factor [48,52] and $I$ is the optical field intensity. For DFB-type lasers the following equation is often used [48]:

$$\Delta v = \left(1 + \alpha^2\right)\frac{\pi h v_0 \Gamma n_{sp}}{2k P_{out}}(\Delta v_c)^2 \tag{2}$$

with $h v_0$ denoting the photon energy, $\Gamma$ the confinement factor, $n_{sp}$ the spontaneous emission factor, $k$ the coupling coefficient, $P_{out}$ the output power and $\Delta v_c$ the full width at half maximum of the cold laser cavity. Equation (2) can be derived from (1) by observing the relations

$$R_{sp} = \Gamma v_g g_{th} n_{sp} \tag{3}$$

$$I = \frac{2P_{out}}{h v_0 v_g \alpha_m} \tag{4}$$

$$g_{th} = \frac{\alpha_m}{k} \tag{5}$$

$$\Delta v_c = \frac{v_g \alpha_m}{2\pi} \tag{6}$$

We calculate the quantities entering (2), obtaining $\alpha \approx 2.6$, $\Gamma = 7.14\%$, $n_{sp} \approx 4.6$, $k = 8$ cm$^{-1}$ and $\Delta v_c \approx 50$ GHz. By using these quantities in (2) to calculate the linewidth, we obtain the theoretical expected dependence of the linewidth on the output power depicted in Figure 11c. The comparison reveals that the variation trend of the FWHW linewidth with the output power measured by the frequency noise PSD is basically consistent with the theoretical limit calculated from (2). The theoretical linewidth rebroadens the FWHM linewidth, which is attributed to the technical noise rather than to the noise of the diode laser [53]. The two linewidths are inversely proportional to the output power. The linewidth successfully narrowed to 30.9952 KHz benefits from the high output power, which is the expected result of a high Q factor. In an effort to obtain a high Q factor, there are several designs employed in this paper. which depends on the designs in this paper. (1) The 400 nm waveguide layer guarantees the minimum free-carrier loss and leakage loss. (2) The grating with a coupling coefficient of 8 cm$^{-1}$ ensures single-mode lasing and a low loss. (3) The grating with four $\lambda/16$ phase shift regions is applied to suppress the spatial hole-burning effect. The carriers used for the side mode gain decrease, and those used for the lasing wavelength gain increase, which is equivalent to a decrease in the loss. (4) The low background doping concentration of the undoped waveguide layers can reduce the free-carrier loss. (5) The dry etching and the wet etching are combined to obtain a smooth and high-consistency surface; such an etching method can reduce loss which generates during growth. (6) The 0.01% reflectivity coating ensures the mode filtered by Bragg mode selection conditions can stably lase. The carriers avoided exhaustion at the side mode gain, which also reduces the loss.

In order to detect the weak magnetic field fluctuations of the human brain, the MEG also requires very low relative intensity noise (RIN) of lasers, especially the noise level at 1 kHz [54,55]. Figure 11d shows the RIN in the frequency range of 0–10 MHz. When the relaxation oscillation frequency is about 1 kHz, the RIN level reaches −144.73 dBc/Hz. The RIN value is inversely proportional to the cube of output power and low RIN can be achieved by the high output power. The high SMSR effectively reduces the high-order mode noise and reduces the RIN. The application of the 8° inclined fiber and the 45 dB bipolar isolator also obviously reduces the RIN via suppressing reflection. The W60 heat sink with low tungsten content is beneficial to tighter connection with the TEC, and conducive to the reduction in the RIN via avoiding the temperature instability caused by welding air gap between the TEC and the heat sink.

The power and frequency reliability stands for long-term stability, which is essential for high-end applications. The detailed power fluctuations over 24 h are illustrated in Figure 12a. The power shift is less than 0.04%, and the corresponding fluctuations may have been caused by the pump source because it had been used for a long time. The pump current source fluctuated by 0.035% across a 24 h period, which was basically consistent with the power fluctuation within 24 h. The result indicates that the laser can work stably and constantly for the MEG. The long-term continuous operation of the DFB laser working at beat frequency, plotted in Figure 12b, has been monitored for 24 h. The change only is 138 MHz. The long-time wavelength stability of the laser satisfys the needs of the MEG.

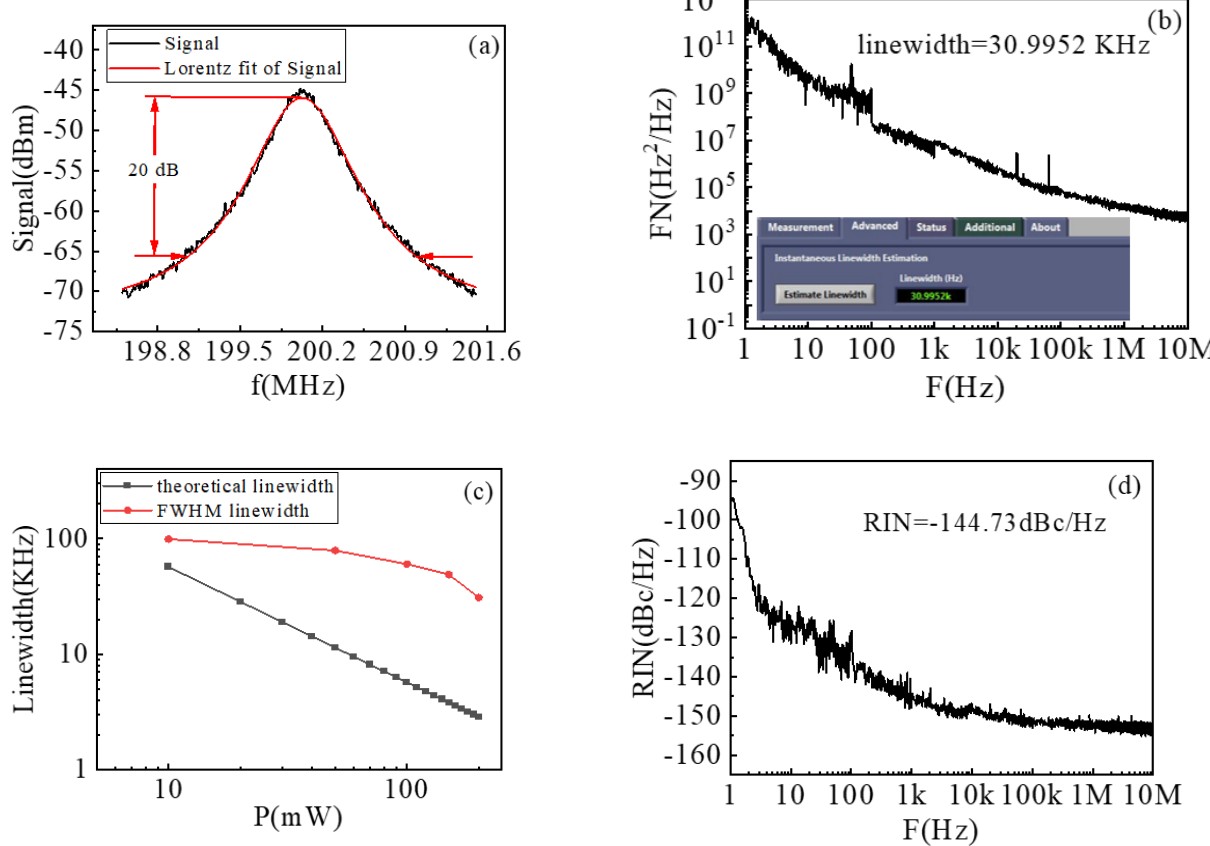

**Figure 11.** (**a**) Linewidth measured by the SDH method (**b**) Linewidth measured by PSD of frequency noise (inset is a test photo) (**c**) FWHM measured by PSD of frequency noise and theoretical linewidth vs. output power (**d**) RIN spectra of the DFB laser in range frequency of 0–10 MHz.

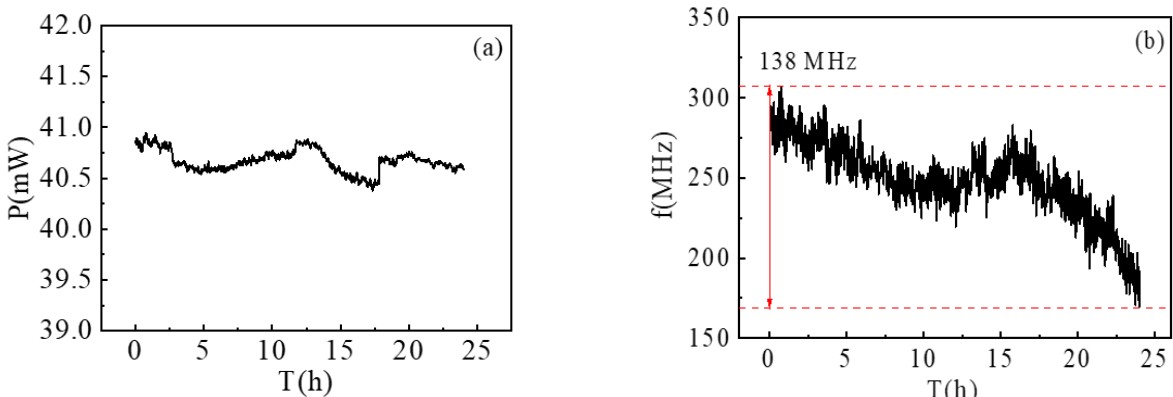

**Figure 12.** (**a**) Output power of the laser at a 1 min interval over 24 h period. (**b**) Beat frequency of two lasers at a 1 min interval over 24 h period.

## 4. Discussion

In this paper, a 1083 nm laser chip is designed. The traditional InGaAs/GaAs quantum well structure that satisfies the requirements of the emission wavelength will have a lattice mismatch greater than 2%, which is the main cause of distortion. Therefore, the innovative InGaAs/GaAsP quantum well structure is proposed to solve the above problem. The waveguide and the grating structure are optimally designed, and the growth environment conditions such as the growth temperature and the proportion of III–V groups are experimentally modified to obtain the expected epitaxial wafers. Furthermore, the

double-trench-ridge waveguide structure is designed to achieve the single-fundamental-mode output and successfully suppress the high-order mode emission. As a result, the output power of the chip can reach 285 mA at 500 mA. The secondary lens package method adopted in this paper also provides a guarantee for the excellent performance of the module. The module output power is 30.74 mW at 100 mA, the linewidth is 30.9952 kHz, and the RIN noise is −144.73 dBc/Hz at 1 kHz. The 1083 nm lasers with a precisely controlled wavelength and a sharply narrow linewidth will promote significant development for the MEG.

**Author Contributions:** Conceptualization, M.W., W.W. and H.Y.; hardware, M.W. and S.L.; validation, M.W., S.L. and H.Y.; writing—original draft preparation, M.W.; writing—review and editing, M.W., W.W. and H.Y.; project administration, J.L. and Y.C. All authors have read and agreed to the published version of the manuscript.

**Funding:** This research was funded by the National Key Research and Development Program of China, grant number 2021YFB2800402.

**Acknowledgments:** The authors are grateful to the State Key Laboratory on Integrated Optoelectronics, Institute of Semiconductors, Chinese Academy of Sciences and School of Integrated Circuits, University of Chinese Academy of Sciences.

**Conflicts of Interest:** The funders had no role in the design of the study; in the collection, analyses or interpretation of data; in the writing of the manuscript; or in the decision to publish the results.

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
