# Peer review of "A 1083 nm Narrow-Linewidth DFB Semiconductor Laser for Quantum Magnetometry"

_photonics, doi:10.3390/photonics10080934_

Round 1
Reviewer 1 Report
This manuscript demonstrated a high power, narrow linewidth DFB laser at 1083nm. The epitaxial design and growth of laser, device preparation and module packaging are completed. At the same time, the power and spectral properties of the DFB laser are tested. The experiment results are solid and the conclusion is supported by the data. However, I do have some questions about the manuscript:
1. In order to understand the device structure more intuitively, the authors preferred the device structure diagram and DFB grating SEM diagram.
2. The reflectivity of the anti-reflection film is difficult to reach 0.01%. Is the value measured experimentally or calculated by simulation?
3. In fig. 8, authors may mark the parameter value of side-mode suppression ratio.
4. At Page 7-Lines 191, authors propose the highest output power of DFB laser is higher than other 1083nm DFB lasers. Please give comparative references.
-
Some paragraphs in the manuscript need to be revised。
Author Response
We would like to thank the reviewer 1 for the valuable suggestions. We have carefully considered the reviewer’s suggestions and revised our manuscript accordingly.
Point 1: In order to understand the device structure more intuitively, the authors preferred the device structure diagram and DFB grating SEM diagram.
Response 1: Thank you very much for your comments. We have added the device structure diagram and DFB grating SEM diagram in the corrected manuscript (Figure 5, on the page 5; Figure 6(a), on the page 6).
Point 2: The reflectivity of the anti-reflection film is difficult to reach 0.01%. Is the value measured experimentally or calculated by simulation?
Response 2: Thank you very much for your question. The value of reflectivity is measured experimentally.
Point 3: In fig. 8, authors may mark the parameter value of side-mode suppression ratio.
Response 3: Thank you very much for your comment. We have marked the parameter value of side-mode suppression ratio (Figure 9(b), on the page 9).
Point 4: At Page 7-Lines 191, authors propose the highest output power of DFB laser is higher than other 1083nm DFB lasers. Please give comparative references.
Response 4: Thank you very much for your comment. We have added comparative references. But the research of 1083nm DFB lasers is few, and comparative references are few. (Lines 196, on the page 8).
Reviewer 2 Report
In this manuscript, the authors present the fabrication and characterization of a DFB 1083 nm laser, which holds potential applications in magnetometry for brain science. The laser is pigtailed in a standard butterfly package, ensuring a compact design. The authors evaluate its performance in terms of output power (285 mW), intensity noise (-144.73 dBc/Hz), linewidth (~31 kHz), and tunability (2.4 nm with thermal tuning). The fabrication process demonstrates novelty, and the laser's performance, particularly the linewidth, is remarkable. Therefore, I recommend publishing this manuscript after addressing the following issues.
1) Why does the output power saturate above 500 mA? The authors should provide an explanation.
2) It would be valuable if the authors could discuss the feasibility of gain switching this laser. A high-power pulsed source at this wavelength, without the need for a fiber amplifier, would be of interest for chemical sensing applications.
3) The author should consider labelling the epi layers in the SEM (figure 5).
4) Why is the theoretical linewidth much narrower than the measured results in Figure 10 (c)? What is the main broadening factor in this laser?
5) Why is 1 kHz noise important for MEG? The author should cite some reference.
Author Response
We would like to thank the reviewer 2 for the valuable suggestions. We have carefully considered the reviewer’s suggestions and revised our manuscript accordingly.
Point 1: Why does the output power saturate above 500 mA? The authors should provided an explanation.
Response 1: Thank you very much for your comments. We have added the explanations in the corrected manuscript (line 199 to 202, on the page 8).
Point 2: It would be valuable if the authors could discuss the feasibility of gain switching this laser. A high-power pulsed source at this wavelength, without the need for a fiber amplifier, would be of interest for chemical sensing applications.
Response 2: Thank you very much for your comments. We have discussed the feasibility of gain switching this laser in the corrected manuscript (line 208 to 213, on the page 8).
Point 3: The author should consider labelling the epi layers in the SEM (figure 5).
Response 3: Thank you very much for your comment. The boundaries between the epi layers in the SEM are not obvious. In order to show the structure more clearly, we have added device structure diagram (Figure 5, on the page 5).
Point 4: Why is the theoretical linewidth much narrower than the measured results in Figure 10 (c)? What is the main broadening factor in this laser?
Response 4: Thank you very much for your questions. Measured linewidth is re-broadened by the noise of the current controller. So theoretical linewidth is much narrower than the measured results. We have added the explanations in the corrected manuscript (line 283 to 285, on the page 11). The main broading factor in this laser is loss introduced during epitaxy growth.
Point 5: Why is 1 kHz noise important for MEG? The author should cite some reference.
Response 5: Thank you very much for your comment. We have added some reference in the corrected manuscript (line 301, on the page 11).
Reviewer 3 Report
Dear Authors,
your paper is very interesting and the described laser is very promising. I have questions.
1. Experimental results in Fig.8(a) have no points of the measurement with error bars. They look like theoretical curve.
2. Spectrum in Fig.8(b) has wave-form backgroung, and you do not explain the reason of it.
3. You do not discuss the stability of fixed parameters (temperature or current) during the experiment.
4. In lines 299-300 you mention that "The power shift is less than 0.04%, and the corresponding fluctuations may have been caused by the pump source because it had been used for a long time." I guess the connection of the shift with the drift of pump source output may be discussed in more detailes. Generally speaking, it would be usefull to present a recuirement on parameters of devices which you used for achivement so impressive characteristics of the laser.
I have only one comment. It would be very interesting to know a variation of the laser parameters under their mass production. In other words, how big is the yield of the lasers with the described parameters within 5 percent of tolerance in mass production?
Author Response
We would like to thank the reviewer 3 for the valuable suggestions. We have carefully considered the reviewer’s suggestions and revised our manuscript accordingly.
Point 1: Experimental results in Fig.8(a) have no points of the measurement with error bars. They look like theoretical curve.
Response 1: Thank you very much for your comments. The performance of the 1083nm lasers is stable, and the PIV curves of repeated measurements are consistent. Therefore, experimental results in Fig.8(a) have no points of the measurement with error bars.
Point 2: Spectrum in Fig.8(b) has wave-form backgroung, and you do not explain the reason of it.
Response 2: Thank you very much for your comments. We have added the reason of wave-form backgroung in the corrected manuscript (line 217 to 219, on the page 8).
Point 3: You do not discuss the stability of fixed parameters (temperature or current) during the experiment.
Response 3: Thank you very much for your comments. We have added the discussion about fixed parameters during experiment in the corrected manuscript (line 228 to 229, on the page 9; line 234 to 236, on the page 9).
Point 4: In lines 299-300 you mention that "The power shift is less than 0.04%, and the corresponding fluctuations may have been caused by the pump source because it had been used for a long time." I guess the connection of the shift with the drift of pump source output may be discussed in more detailes. Generally speaking, it would be usefull to present a recuirement on parameters of devices which you used for achivement so impressive characteristics of the laser.
Response 4: Thank you very much for your comments. We have added the detials in the corrected manuscript (line 317 to 319, on the page 12).
I have only one comment. It would be very interesting to know a variation of the laser parameters under their mass production. In other words, how big is the yield of the lasers with the described parameters within 5 percent of tolerance in mass production?
Response 5: Thank you very much for your comments. In our small batch experiment, 50% is the yield of the lasers with the described parameters within 5 percent of tolerance.